# The Critical Role of Stereotactic Body Radiation Therapy in Multimodal Treatment of Lung Metastasis from Bone and Soft Tissue Sarcomas

**DOI:** 10.3390/cancers16213593

**Published:** 2024-10-24

**Authors:** Alessandra Longhi, Andrea Marrari, Cecilia Tetta, Anna Parmeggiani, Orlando Parise, Cristina Ferrari, Fabrizio Salvi, Giovanni Frezza

**Affiliations:** 1Struttura Complessa Osteoncologia, Istituto Ortopedico Rizzoli, Istituto di Ricovero e Cura a Carattere Scientifico (IRCCS), 40136 Bologna, Italy; andrea.marrari@ior.it; 2Cardiovascular Research Institute Maastricht (CARIM), Maastricht University, Universiteitssingel 50, 6229 ER Maastricht, The Netherlands; cinilia69@gmail.com (C.T.); o.parise@icloud.com (O.P.); 3Dipartimento di Diagnostica per Immagini, Osp Ceccarini, 47838 Riccione, Italy; anna.parmeggiani@auslromagna.it; 4Department of Cardiac Surgery, Universitair Ziekenhuis Brussel, Laarbeeklaan 101, 1090 Brussels, Belgium; 5Laboratorio di Oncologia Sperimentale, Istituto Ortopedico Rizzoli, Istituto di Ricovero e Cura a Carattere Scientifico (IRCCS), 40136 Bologna, Italy; cristina.ferrari@ior.it; 6Unita’Operativa di Radioterapia, Ospedale Bellaria, Via Altura 3, 40139 Bologna, Italy; f.salvi@ausl.bologna.it (F.S.); g.frezza54@gmail.com (G.F.)

**Keywords:** lung metastasis, stereotactic body radiation therapy, bone sarcoma, soft tissue sarcoma

## Abstract

Surgical metastasectomy has been the primary choice for local therapy in patients with lung metastasis secondary to sarcoma. Over recent decades, however, stereotactic body radiation therapy (SBRT) has been employed more extensively in the treatment of lung metastasis. SBRT is associated with less toxicity and similar outcomes compared with surgical metastasectomy. When used with chemotherapy and surgery in a multimodal treatment plan, SBRT can improve the cure rate and prolong survival.

## 1. Introduction

Sarcomas are a group of rare and heterogeneous solid tumors of mesenchymal origin that account for 1% of adult and 15% of childhood malignancies [1,2]. Despite adequate multimodal treatment, the risk of relapse at 5 years is 40% for soft tissue sarcoma (STS) and 30–35% for bone sarcoma (BS) [1,2,3]. Both forms commonly metastasize to the lungs [4,5,6]. The treatment of lung metastasis (LM) varies by histotype, grade, diagnostic timing (synchronous or metachronous pulmonary lesions), number and resectability, interval to recurrence, presence of extrapulmonary disease, and preoperative response to chemotherapy. Pulmonary metastasectomy is usually chosen for isolated LM and demonstrates excellent local control and survival [7,8]. However, new lung nodules occur in up to 50% of patients despite aggressive surgical resection [9], and patients may only be eligible for palliative chemotherapy in the presence of multiple lung and extrapulmonary metastases. Furthermore, given the limited chemosensitivity and indolent nature of some histotypes, local treatments may have a role in prolonging survival.

Stereotactic body radiation therapy (SBRT), which is an external beam radiotherapy technique, allows the delivery of very high doses of radiation in single or few fractions [10]. The technique has been widely and successfully employed in cancer care both for primary and metastatic disease [11], proving to be safe and effective in older or fragile patients and in those with oligometastatic disease [12]. The randomized COMET trial in oligometastatic patients with 1–5 metastatic lesions from different primary cancers (breast ca, colon ca, prostate ca), with all metastases amenable to SBRT, resulted in an improved overall survival in the arm of patients treated with SBRT vs. those treated with a palliative standard of care [13]. Retrospective and prospective studies of SBRT in STS and BS have suggested excellent local control with limited toxicity [14,15,16,17], although no prospective trials have compared SBRT with metastasectomy. One recent systematic review comparing SBRT and surgical metastasectomy for LM in patients with sarcoma concluded that SBRT was associated with higher survival, even when reserved for unfit patients at risk of unfavorable outcomes [18]. However, in the absence of data from randomized clinical trials, these data need to be interpreted with caution. We studied the outcomes of patients with LM due to STS and BS and treated them with a multimodal approach, focusing our result on SBRT treatments.

## 2. Materials and Methods

### 2.1. Study Design, Setting, and Endpoints

This retrospective study reports on the outcomes of SBRT for LM in patients with STS and BS at two institutions in Bologna, Italy: the Rizzoli Orthopedic Institute and Bellaria Hospital Radiotherapy service. After Ethical Committee approval (CE/AVEC no. 15867; approved on 28 October 2021), we reviewed the charts of patients enrolled between 2010 and 2021, including follow-up date to December 2023. The study was conducted according to national law regulating the conduct of observational studies (no. 11960, 13 July 2004).

The primary endpoints were the irradiated nodule response, relapse free interval of irradiated nodules at 1 and 2 years, and overall survival (OS) from the first SBRT. Secondary endpoints were the distant LM-free interval from first SBRT to next relapse or last follow-up, other secondary end points were OS from first LM event to last follow-up or death, prognostic factors affecting multimodal treatments, and SBRT toxicity.

### 2.2. Inclusion Criteria

We enrolled patients with a diagnosis of primary STS or BS who developed LM (or who had synchronous LM at diagnosis), as long as they had received at least one course of SBRT and had new-onset pulmonary nodules during follow-up. Patients could have received other treatments for LM before SBRT, including metastasectomy and chemotherapy. In patients with localized disease at primary tumor diagnosis, disease-free interval (DFI) 0 was the period from the date of diagnosis to first lung relapse. In patients with LM at primary tumor diagnosis, DFI 0 was the time from first disease treatment to first LM progression. If complete remission occurred after treatment for relapse, oncologic assessment by chest computed tomography (CT) was planned every 2–3 months and new-onset nodules with a diameter >5 mm that grew in subsequent chest CT scans performed after 2–3 months were deemed metastatic. At first LM diagnosis, nodules were classified by laterality (unilateral or bilateral) and number (A, B, or C, representing 1–3, 4–9, and ≥10 nodules, respectively). Relapse sites and dates of progression, treatment, and extrapulmonary relapse were recorded. Responses to SBRT and the occurrence of post-SBRT adverse events were evaluated using the RECIST criteria [18] and CTCAE 5.0 criteria [19], respectively. DFI 1 was the interval from the start date of treatment (surgery, chemotherapy, SBRT) for the first LM relapse to the date of a second relapse; DFI 2 was that from the second to the third relapse; and DFI 3 was that from the third to the fourth relapse.

### 2.3. Treatment with SBRT

Patients suitable for SBRT were selected by multidisciplinary discussion, received information about the treatment and possible adverse events, and gave written consent. A vacuum pillow was used to immobilize patients in the supine position, while abdominal compression was used to reduce diaphragm movement. Before 2016, CT scanning used a slow rotation time (3 s), as described by Lagerwaard et al., to define the target position accurately during respiratory phases [20]. Since 2016, we have used four-dimensional CT and defined the gross tumor volume in serial images. The internal target volume was the total of all gross tumor volumes. Patients with multiple or highly mobile lesions were treated in deep inspiration breath hold (DIBH) using Active Breath Control (ABC^©^; Elekta, Bologna, Italy) or Catalyst HD^©^ (C-RAD, Berlin, Germany) devices. We recorded the planning tumor volume (PTV), number of nodules treated, radiotherapy dose per nodule, and fractions per session. The PTV was the internal target volume plus 3–5 mm and 6–9 mm margins in the transverse and superior-inferior planes, respectively. An 80% isodose was prescribed to cover >98% of the PTV, with administered doses ranging from 30 to 60 Gy in 3–8 daily fractions. Central lesions typically required more fractions than peripheral lesions. Patients underwent three- dimensional conformal radiotherapy with 8–10 non-coplanar fields or with volumetric-modulated arctherapy. Dose constraints to organs at risk have been described [14].

### 2.4. Statistical Analysis

Continuous data were tested for normality by the Shapiro–Wilk test and expressed as means and standard deviations or medians and interquartile ranges (IQR) or ranges. Discrete data are expressed as counts and percentages. Analysis of variance tested the means of continuous variable between groups. Survival analysis was performed using the Kaplan–Meier method and log-rank test to assess statistical differences between the STS and BS groups. OS was considered from the first metastatic event or SBRT session to the last visit or death. This choice to evaluate both OS from first LM or first SBRT session was due to the fact that 1st-course SBRT was not usually the first treatment at 1st LM relapse, and different other treatments (surgery, chemotherapy) were administrated at different relapses that occurred in the patient’s history. Therefore, OS from the first metastatic event could better express the value of a multimodal treatment. Univariable and multivariable Cox models were employed to determine the predictors of mortality after SBRT. Variables with a *p*-value < 0.1 or with clinical relevance were entered into multivariable Cox regression analysis. The Kaplan–Meier method was also used to assess progression-free survival for treated nodules and for LM, with the latter considering the time interval from first SBRT to next LM relapse or follow-up. Statistical analysis was performed in R (version 4.3.3; R Foundation for Statistical Computing, Vienna, Austria), Python (version 3.10.13; Python Software Foundation), and Jasp (version 0.18.3; Jasp Team). *p*-values < 0.05 were considered statistically significant.

## 3. Results

### 3.1. Participant Characteristics

After reviewing all cases of sarcoma during the study period, 102 patients with LM were eligible (51 each had BS and STS). Twelve patients were excluded those who received RT to thorax for palliative purpose [4] or for inadequate follow-up data [8]. Their median age was 40 years (range, 11–81), 31 were female and 71 were male, and the median follow-up was 4.8 years (95% CI, 4.1–7.5). At primary tumor diagnosis 76 (74.5%) had localized disease and 26 (25.5%) had metastatic disease (25 had LM and 1 had bone metastasis). Patient characteristics are reported in Table 1 and the BS and STS histotypes are reported in Table 2. There were 51 STS and 51 BS, and among BS, there were 18 Ewing sarcoma and 33 non-Ewing sarcoma, of which the majority were osteosarcoma (21 pts). Most patients underwent surgery for their primary tumor, except for 8 with Ewing sarcoma and 1 with liposarcoma who received local radiotherapy. In patients with localized disease, the median interval from primary tumor diagnosis to LM was 21 months (range, 2–282). The number of nodules at first LM recurrence was 0–3 (66 pts), 4–9 (21 pts), and ≥10 (15 pts). The median number of Recurrences episodes was 4 per patient (range, 1–12) Figure 1. First disease recurrence was LM in 71 patients (70%), local relapse in 16 (15.8%), and local relapse together with LM in 7 (6.9%). Patients with local relapse before LM were evaluated from the time of first LM. At study entry, 46 patients had unilateral LM and 56 had bilateral LM. Furthermore, 75 patients had already undergone chemotherapy and 52 had already undergone surgery for LM. All 102 patients received SBRT at least once, with a median of 2 (range, 1–11) SBRT courses per patient. Surgery was employed more at first relapses and SBRT was used more in advanced relapses. Relapse treatments are summarized in Table 3, in which we report single treatment and SBRT plus chemotherapy frequency at each relapse The numbers of bilateral LMs and extrapulmonary metastases increased with subsequent relapses. Extrapulmonary relapse was local in 14, bone in 12, brain in 6, soft tissue in 4, soft tissue and bone in 4, liver in 2, and other sites in 12. Lung progressive disease after surgery was defined by the appearance of at least one new nodule or more nodules of at least 5 mm or growing nodules in subsequent chest CT after 2–3 months and that was deemed as metastatic by multidisciplinary team.

### 3.2. Response to SBRT

Overall, 276 nodules were treated with SBRT.

The median nodule diameter was 10 mm (IQR, 6.7–15.2) maximum diameter was 5 cm, and the median dose was 48 Gy per nodule (IQR, 40–54 Gy). In patients with multiple nodules each SBRT course treated 1–4 nodules maximum (3 all together at maximum if homolateral). In case of multiple lesions total dose was sometime reduced in order to preserve dose limits to organs at risk. Generally, in centrally located lesions total dose was reduced (40 Gy in 5 fractions) or split into a greater number of fractions (54 Gy in 8 fractions) to reduce the risk of late complications. According to the RECIST criteria [17], we observed complete remission in 168 (61.3%), partial response in 18 (6.6%), stable disease in 24 (8.8%), and progression in 61 (22%). We could not evaluate three patients. After the first SBRT course, the progression-free rates of irradiated nodules were 86% at 1 year (95% CI, 81–90) and 78% at 2 years (95% CI, 72–83) Figure 2, while the median survival from first SBRT treatment was 2.9 years (95% CI, 2.2–3.9) and the 5-year OS from first SBRT (principal end point) was 37% (95% CI, 28–49) Figure 3.

Median distant LM-free interval from first SBRT to next relapse was 6 months (range, 5–10). After the first SBRT, 34% (95% CI, 26–45) and 20% (95% CI, 14–30) were free of distant LM at 1 and 2 years, respectively. The median DFI decreased after each relapse: DFI 1 was 8 months (range, 0–120), DFI 2 was 6.5 months (range, 1–80), and DFI 3 was 5 months (range, 0–118).

On 31 December 2023, there had been 60 deaths; among the 40 survivors, 22 were alive with disease and 20 were alive without disease. Overall, median survival from first LM appearance or progression was 4.8 years (95% CI, 4.1–7.5); for BS and STS, median survival was 5.7 years (95% CI, 4.1–8.9) and 4.8 years (95% CI, 4.1–9.9), respectively. The 5-year OS for all groups from first metastasis or from metastatic progression (if LM already present) was 49% (95% CI, 39–60); for BS and STS, the 5-year OS was 52% (95% CI, 39–69%) and 46% (95% CI, 33–63%), respectively (no statistically significant difference. No statistically significant difference in objective response and overal survival from first LM was noted between STS and BS *p* value = 0.63 (Figure 4). As well as inside BS no statistically significant Objective Response and Overall Survival was noted between Ewing sarcoma (18 pts) and other bone sarcomas (33 pts) *p*-value log-rank test 0.24 (Figure 5). We did not observe any difference in local control of LM according to different techniques, but in more recent years, the use of DIBH (deep inspiration breath hold) allowed, when possible, for the easier reduction of the total dose of radiation to the lungs, especially in patients with multiple lesions.

In patients with localized disease, the number of nodules (1–3, 4–9, >10) was not significantly correlated with the interval between primary tumor diagnosis and metastatic relapse. The maximum lung nodule diameter was also not associated with response to SBRT. However, multivariable analysis revealed that receiving chemotherapy before SBRT was associated with worse survival and that having a longer interval from the first SBRT to the next recurrence or last visit was associated with a better prognosis. The DFI 1 interval was also significantly affected by treatment modality (Kruskal–Wallis test, *p* = 0.003). Compared with chemotherapy, pairwise comparisons (Dunn’s post hoc test) showed a significant increase in the DFI when using surgery (*p* = 0.008) or SBRT (*p* = 0.026) Figure 6. Complications of SBRT were typically mild and included 4 rib fractures, 3 pneumothoraxes, and 23 cases of post-radiotherapy pneumonitis (grades 1–2), of which 22 had grade 1 pleural effusion.

### 3.3. Analysis of 20 Long-Term Survivors

Long-term survivors in cancer patients were defined as a survival of 5 years or longer from diagnosis of primary tumor [21]. Table 4 summarizes the details of 20 patients (19.6%) who survived without evidence of ongoing disease: 9 had BS, 12 had STS, 2 had metastasis at diagnosis (1 osteosarcoma and 1 STS). The median time from the first SBRT to the last visit was 4.6 years (range, 3.2–6.7), while the median interval from first metastasis to last follow-up was 7.4 years (IQR, 4.8–9.4; range, 2.4–19.4 years). The median DFI 0 and DFI 1 were 12 months (range, 4.75–26.5) and 11.5 months (range, 3.75–41.75), respectively.

## 4. Discussion

We confirmed that chemotherapy before SBRT is an unfavorable prognostic factor. This is consistent with evidence reported by Lindsay et al. [22] and probably reflects selection bias (i.e., patients who received chemotherapy at earlier relapses had more advanced disease unsuitable for local LM treatments. Furthermore, the present study showed that surgery for LM was more frequently used at the first relapse, while SBRT was preferred from the second relapse onward. Concerning relapse intervals, surgery and SBRT both offered prognostic benefits compared with chemotherapy alone or in combination with chemotherapy or surgery. Comparison can be made with reports from before the extensive employment of SBRT for LM, when the 5-year OS of STS was 30–40% after the surgical treatment of relapsed metastatic LM [7]. In osteosarcoma with metachronous metastases, the 5-year OS post-metastasectomy was 28–38% [23,24], reaching 50% for those with a single LM, while in Ewing sarcoma, the 5-year OS after lung recurrence was around 20–30% [25,26]. A pre-SBRT report identified long-term survival in just 16.8% of 110 relapsed osteosarcomas [27]. Finally, in a study on 436 patients with STS (including G1) only 9% survived to 5 years after the diagnosis of metastasis [28]. A recent paper reported 66 pts with LM from sarcoma, 93% soft tissue sarcoma, 7% osteosarcomas, with a median follow up of 36 ms from time to first SBR reported a 12 and 24 ms overall survival was 74% (95% CI 64–86%) and 49% (38–63%), respectively. The 24-month cumulative incidence of local failure was 7.4% [29]. Our study included only patients who received at least one course of SBRT with or without other treatments (e.g., surgery or chemotherapy) for different relapses. Our data show a longer 5-year OS of 49%, with 20 of the 102 patients surviving long-term without disease (median interval from first metastatic event to last follow up, 7.4 years; range, 2.4–19.4 years). Moreover, this research benefits from a larger sample size and longer follow-up, despite showing similar irradiated nodule control, when compared with existing research (Table 5) [15,29,30,31,32].

However, our findings are limited by the retrospective design. No precise algorithm exists to guide decisions about treatment options for LM. In our practice, when extensive surgery is required for central LM (e.g., lobectomy or pneumonectomy), we prefer SBRT because it has low invasiveness, is associated with less acute toxicity, can be administrated multiple times (one patient received 11 successive SBRT treatments), and is less expensive than surgery [33,34] but on the other side we are aware that surgery after SBRT treatment can be more difficult to perform. SBRT is also a preferable option for unfit or elderly patients, as well as in oligoprogressive disease [35]. We prefer surgery for young, fit patients and when a histology assessment is requested for doubt nodules, but recently SBRT is going to be administrated more frequently also in younger and fitter patients. Although randomized trials on SBRT in multimodal treatment for LM are difficult to perform, the ongoing OligoRARE trial (NCT 04498767) is seeking to evaluate the value of SBRT added to other treatments in patients with different cancers.

## 5. Conclusions

In conclusion, SBRT for LM secondary to STS and BS is an effective alternative to surgery. It is an additional tool that can be used alongside chemotherapy and/or surgery to improve cure rates (i.e., multimodal treatments appear to be able to cure approximately 1 in 5 patients), to treat more relapses (i.e., SBRT can be administrated several times), and ultimately, prolong survival after relapse.

## Figures and Tables

**Figure 1 cancers-16-03593-f001:**
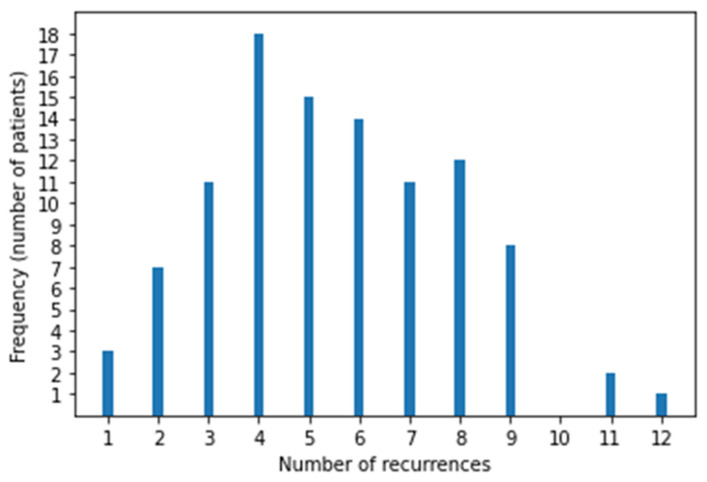
Distribution of recurrences in 102 patients.

**Figure 2 cancers-16-03593-f002:**
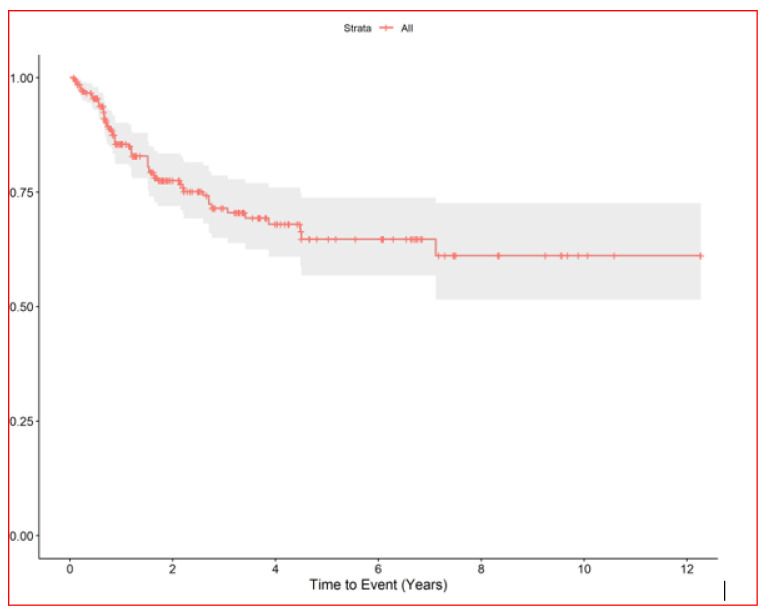
RELAPSE FREE INTERVAL of irradiated lung nodules.

**Figure 3 cancers-16-03593-f003:**
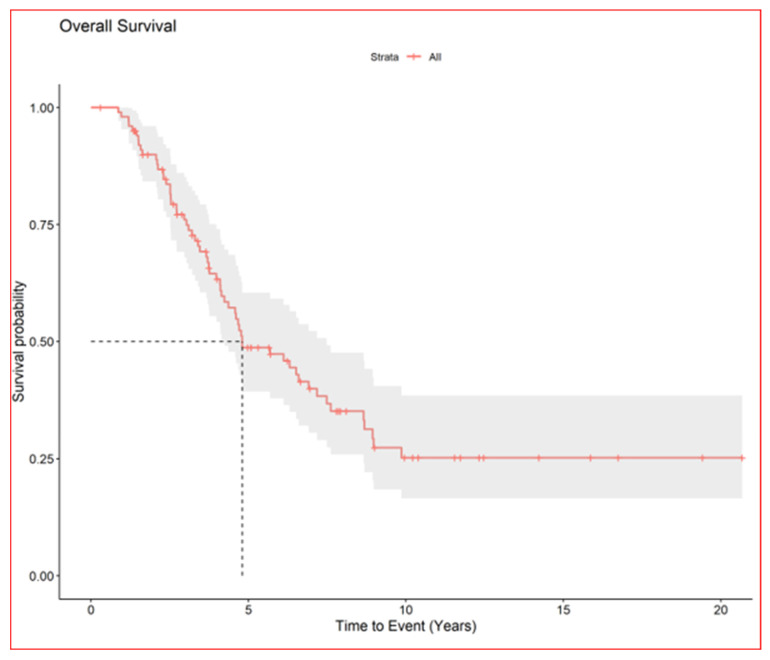
Overall survival of all patients.

**Figure 4 cancers-16-03593-f004:**
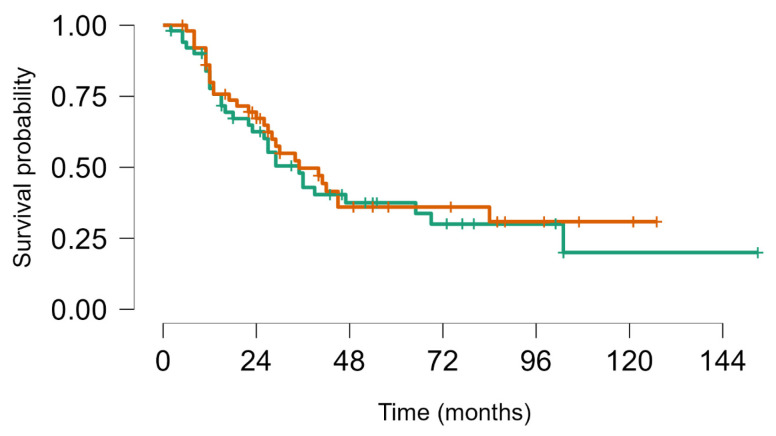
Survival from first metastatic episode to last FUP: Soft tissue sarcoma (brown) vs. bone sarcoma (green), Log rank test *p* value = 0.63.

**Figure 5 cancers-16-03593-f005:**
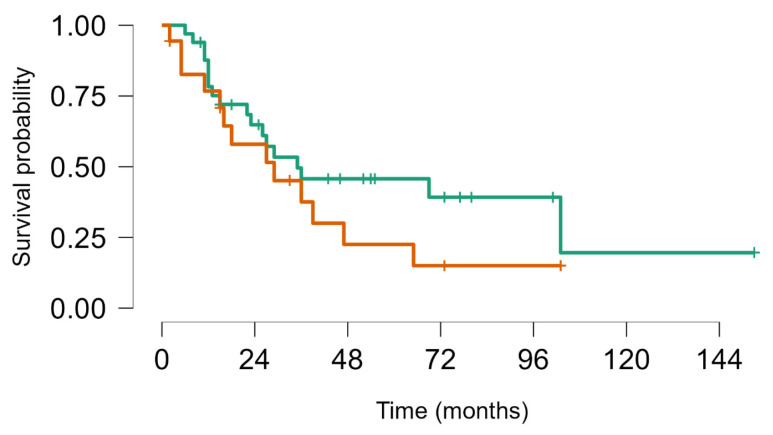
Survival from first lung metastasis to last FUP Ewing sarcoma (brown) vs. other BS (green). Log rank test *p* value = 0.24.

**Figure 6 cancers-16-03593-f006:**
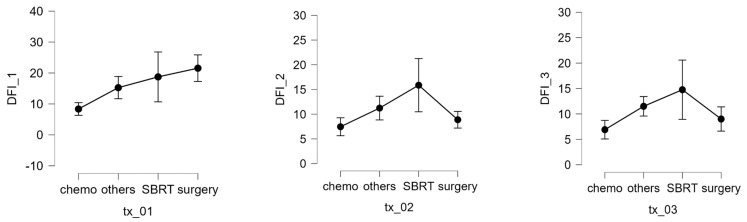
DFI 1, 2, and 3 expressed in months and different therapy.

**Table 1 cancers-16-03593-t001:** Patient data.

Parameters		
Age	Median	40 (11–81) year
Sex	Male	71
	Female	31
Histotype	Bone Sarcoma	51
	Soft Tissue Sarcoma	51
Stage at Dx	Localized	76
	Metastatic	26
Patients with lung metastatic nodules at 1st relapse	1–3	66
	4–9	21
	≥10	15
Recurrences per patient, n	Median (range)	4 (1–12)
Pre-SBRT chemo, n patients		75
Pre-SBRT surgery, n patients		52
N irradiated nodules		276
Median Nodule diameter		10 (IQR 6.7–15.2) mm
Median dose Gy		48 (40–54)
Number of SBRT course for each patient	Median	2 (1–11)
RECIST Response, %	CR	61.3%
	PR	6.6%
	SD	8.8%
	PD	22%
	NE	1.1%
Local control rate after 1st SBRT	12 months	86%
	24 months	78%
Survival from 1st SBRT	Median OS	2.9 (95% CI, 2.2–3.8)
	5-year OS	37% (95% CI, 27–48)
Survival from 1st metastases	Median	4.8 years (95% CI, 4.1–7.5)
	5-year	49% (95% CI, 39–60)

Abbreviations: CR, Complete response; PD, progressive disease; PR, partial response; SD, stable disease; Dx, diagnosis; NE, not evaluable; OS, overall survival SBRT, stereotactic body radiation therapy.

**Table 2 cancers-16-03593-t002:** Histotypes.

Histotypes	N
Bone Sarcoma, total	51
Osteosarcoma	21
Ewing Sarcoma	18
Chondrosarcoma	5
Undifferentiated Sarcoma	3
Dedifferentiated Chondrosarcoma	1
Giant Cell Tumor	1
Leiomyosarcoma of bone	1
Mesenchymal chondrosarcoma of bone	1
Soft Tissue Sarcoma, total	51
Undifferentiated Sarcoma	14
Synovial Sarcoma	7
Extra-skeletal Myxoid Chondrosarcoma	5
Leiomyosarcoma	5
Liposarcoma	5
Extra-skeletal Osteosarcoma	4
MPNST	4
Myxofibrosarcoma	2
Triton Tumor	2
ASPS	1
Malignant Myoepithelioma	1
Solitary Malignant Fibrous Tumor	1

Abbreviations: ASPS, Alveolar Soft-Part Sarcoma; MPNST, Malignant Peripheral Nerve Sheat Tumor.

**Table 3 cancers-16-03593-t003:** Relapses and treatments.

Relapse No.	N pt	Surgery	CT	SBRT	SBRT Combo	Other Tx/WS
1st	102	34	30	8	11	19
2nd	99	16	35	15	19	14
3rd	92	12	37	22	17	4
4th	77	9	30	15	12	11
5th	61	5	24	13	13	6
6th	44	2	22	8	3	9
7th	33	2	18	3	3	7
8th	21	0	10	8	0	3
9th	10	0	5	2	0	3
10th	3	0	2	1	0	0
11th	3	0	0	2	0	1
12th	1	0	0	1	0	0

Abbreviations: CT, Chemotherapy; SBRT, stereotactic body radiation therapy; SBRT combo = stereotactic body radiation therapy plus Chemotherapy. Other = other therapies (Surgery + CT combined) or other (i.e., wait and see).

**Table 4 cancers-16-03593-t004:** Data of 20 patients with no evidence of disease and long survival.

Age	Histo	Stage at Dx	N LM	Relapse n 1	R2	R3	R4	R5	R6	RR7	R8	R9	R10	R11	R12	PRDFS ms	Outcome
71	STS	Local	A	S	SBRT	S										51	NED
34	BS	Local	A	S	CT+S	SBRT	CT	SBRT	SBRT	SBRT						120	NED
27	BS	Local	C	S	SBRT											136	NED
40	STS	Metastatic	A	CT	CT +S+SBRT											38	NED
68	STS	Metastatic	A	CT	S x RL + SBRT	SBRT										44	NED
53	STS	Local	A	S	SBRT											240	NED
47	BS	Local	A	S	S	S	S	CT + SBRT	S	S	SBRT	SBRT	SBRT	SBRT	SBRT	205	NED
58	STS	Local	A	Ct	CT	CT	SBRT									100	NED
35	BS	Local	A	S	S	S	S	S	S	S	SBRT	SBRT				191	NED
62	STS	Local	A	Ct	CT	CT+S	S	SBRT								134	NED
59	STS	Local	A	Sbrt												88	NED
15	BS	Local	A	S	CT+SBRT	S										92	NED
61	STS	Local	A	S	S	S	S	SBRT								272	NED
48	STS	Metastatic	A	CT+S	SBRT	SBRT										139	NED
18	STS	Local	B	CT+SBRT	S	S	S									50	NED
21	BS	Metastatic	B	CT+S+SBRT												60	NED
18	BS	Local	A	CT	CT	SBRT										100	NED
17	BS	Local	C	S	CT	CT	S+SBRT									70	NED
70	STS	Local	A	CT+SBRT												116	NED
34	BS	Local	A	CT+S	CT+S+SBRT											94	NED

Abbreviations: BS, Bone sarcoma; CT, chemotherapy; Dx, diagnosis of Primary tumor; nLM, number of lung metastases (A = 1–3 nodules, B = 4–9 nodules, C > 10 nodules); PR-DFS, Post-relapse disease-free survival; S, surgery; SBRT, Stereotactic Body Radiation Therapy; STS, soft tissue sarcoma; R, relapse; NED, Non Evidence of Disease.

**Table 5 cancers-16-03593-t005:** SBRT in lung metastases in sarcoma.

Author (Year) Ref.	No. Patients	Local Control	FU, Median	OS
Lindsay (2018) [22]	44	1-year, 95%	14 months	50% (5 years)
Baumann (2020) [31]	44	1-year, 96%	16 months	43% (2 years)
Navarria (2015) [15]	28	5-year, 96%	65 months	60% (5 years)
Gutik (2023) [30]	18	2-year, 96%	NR	74% (2 years)
Lebow(2023) [29]	66	2 years 92%	36 months	49% (2 years)
Longhi * present	102	1-year, 86%	56 months	49% (5 years)

Abbreviations: FU, follow-up; OS, overall survival. * Current study.

## Data Availability

Data are available from the first author upon a reasonable request.

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
