# Peer review of "The Critical Role of Stereotactic Body Radiation Therapy in Multimodal Treatment of Lung Metastasis from Bone and Soft Tissue Sarcomas"

_cancers, 2024, doi:10.3390/cancers16213593_

Round 1

Reviewer 1 Report

Comments and Suggestions for Authors

Even though this study suffers from all limitations of retrospective analyses, a long follow-up, reasonable size group, and relative rarity of these clinical indications for SBRT makes this study clinically relevant and I read it with interest.

We may expect that SBRT is a valuable treatment option for recurring and may be even for first relapses in lung of BS and STS, however, it is also good to document this, as authors did. Authors showed a remarkable response rate, with good long-term local control of treated LM.

Obviously, like in primary early lung cancer, it will be probably unfeasible to perform a prospective randomized study to compare outcome of surgery vs. SBRT for LM from sarcoma. But this study shows that such an option should be always taken in mind.

I have a few suggestions that may improve this manuscript:

1.   Section of Results should be better presented, in the current form, probably due to formatting problem, some parts of the text appear as the legends to tables, and this is a little difficult to track a content.

2.   In the paragraph 2.4 it is stated “OS was considered from the first metastatic event or SBRT session to the last visit or death”. It is unclear, moreover, OS was considered as one of the primary endpoints, and in the study design was defined as: “overall survival (OS) from first SBRT”. The latter seems more appropriate, because the value of SBRT for LM was evaluated in this study.

3.   This study groups together small cell sarcomas with radioresistant sarcomas. We would be prone the former send for SBRT, whilst the latter for surgery in case of LM. Have you seen any difference for these two groups of sarcomas? This should be at least discussed.

4.   What was the maximum diameter of LM treated with SBRT? What was the policy for large tumors?

5.   How many LM where treated in one SBRT session?

If many, how dose constraints were modified?

Author Response

I have a few suggestions that may improve this manuscript:

  1. Section of Results should be better presented, in the current form, probably due to formatting problem, some parts of the text appear as the legends to tables, and this is a little difficult to track a content. Results data presentation was  improved
  2. In the paragraph 2.4 it is stated “OS was considered from the first metastatic event or SBRT session to the last visit or death”. It is unclear, moreover, OS was considered as one of the primary endpoints, and in the study design was defined as: “overall survival (OS) from first SBRT”. The latter seems more appropriate, because the value of SBRT for LM was evaluated in this study. The primary end point was Overall survival from first SBRT session to last visit or death but we also  calculated the overall survival from first metastatic event to last follow up  This choice to evaluate both OS  is due to the fact that SBRT 1st course was not usually the first treatment  at 1st LM  relapse   and different other treatments (surgery, chemotherapy) were administrated  at different relapse  of the patients history,  so  OS  from first metastatic event  to last follow up can better express the value of a multimodal treatment. this sentence was included in Statistical analysis
  3. This study groups together small cell sarcomas with radioresistant sarcomas. We would be prone the former send for SBRT, whilst the latter for surgery in case of LM. Have you seen any difference for these two groups of sarcomas? This should be at least discussed.No statistical significant difference in response to SBRTwas noted between  small cell sarcoma (18 Ewing sarcoma) and the other 33  bone sarcoma more radioresistant like osteosarcoma (21 pts) ,  as response rate and neither as overall survival as shown in two new added Figures (Fig 5 and 6) .
  4. What was the maximum diameter of LM treated with SBRT?   What was the policy for large tumors? we treat lesions of maximum diameter  of 5 cm Generally in centrally located lesions total dose was reduced (40 Gy in 5 fractions) or splitted in a greater number of fractions (56 Gy in 8 fractions) to reduce the risk of late complications.
  5. How many LM where treated in one SBRT session? Maximum in one session of SBRT we treat 3 lesion at the same time homolateral

6 If many, how dose constraints were modified? . In case of multiple lesions total dose was sometime reduced in order to preserve dose limits to organs at risk.

Reviewer 2 Report

Comments and Suggestions for Authors

This is a large series of lung nodules from sarcoma treated with SBRT and interestingly the local control is very high and even long term survivors. I have a series of questions: how many patients were excluded from the current analysis and the reasons, how did you define a local relapse after surgery; In your series of patients they have been treated with different techniques related to the time period. Did you try to analyze if those different techniques let to different results or did you see any difference depending on the size or type of sarcoma. In your treatment prescription, how did you adapt i for centrally located tumors?

Thank you for your work

Author Response

This is a large series of lung nodules from sarcoma treated with SBRT and interestingly the local control is very high and even long term survivors. I have a series of questions:

1)how many patients were excluded from the current analysis and the reasons,  12 patients were excluded due to inadequate follow up (8)  or because receive SBRT with  palliative purpose (4)

2)how did you define a local relapse after surgery;  Local Relapse was defined as relapse at the site of primary tumor

3) In your series of patients they have been treated with different techniques related to the time period. Did  you try to analyze if those different techniques let to different results or did you see any difference depending on the size or type of sarcoma.  We did not observe any difference in local control of LM according to different techniques employed  during different period of time , but in more recent years the use of DIBH ( deep inspiration breath hold )allowed, when possible, to reduce more easily the total dose to lungs, especially in patients with multiple lesions

4)In your treatment prescription, how did you adapt i for centrally located tumors? Generally in centrally located lesions total dose was reduced (40 Gy in 5 fractions) or splitted in a greater number of fractions (54 Gy in 8 fractions) to reduce the risk of late complications

Reviewer 3 Report

Comments and Suggestions for Authors

Thank you for asking me to review this paper regarding the role of SBRT in the treatment of lung metastases secondary to bone and soft tissue sarcomas.  I have a few minor comments below but overall I think this is a well written paper.

Abstract

This is well written.

Introduction

This section provides a nice background to the study. The authors might consider adding in some figures to show that for all types of sarcoma the prognosis is generally poor once lung metastases have occurred but that here are some patients who can do well with metastasis directed therapy.

The third paragraph starting SBRT would also benefit from referencing the randomised COMET trial data for SBRT which has shown an improvement in overall survival with SBRT albeit in a number of tumour types and in small numbers.

Materials and Methods

This is well written.

Please could the authors explain their SBRT technique in more detail. Did they include patients with central or ultracentral metastases?

Please could the authors clarify why they used OS as either time from metastatic event or SBRT session. Would it not effect the results to use different time points for different patients?

Results

Please could the authors address the issue of surgery beiong used more at first relapse? Is this patient selection going to effect the results and could potentially more patients benefit from SBRT than shown here if some of the younger, fitter patients had SBRT rather than surgery initially?

Please could the authors explain table 3 a little more as it took me some time to make sense of it. What does the other column refer to?

Figure 2 needs labelling.

Discussion

This is a good summary of the current use of SBRT for sarcoma and is well written.  Could the authors comment on the histological subtypes treated and whether they think SBRT is an option for all types of sarcoma, including those which are felt to be more inherently radioresistant such as MPNST or Triton tumours. 

Author Response

This is well written.

Introduction

This section provides a nice background to the study. The authors might consider adding in some figures to show that for all types of sarcoma the prognosis is generally poor once lung metastases have occurred but that here are some patients who can do well with metastasis directed therapy.

We added  Figures  with curves of survival from first LM to last FUP in STSvs BS ( fig 5) and Ewing’s sarcoma vs other BS (fig 6)  with respective p value ( not significant)

The third paragraph starting SBRT would also benefit from referencing the randomised COMET trial data for SBRT which has shown an improvement in overall survival with SBRT albeit in a number of tumour types and in small numbers. COMET SABR study was mentioned in Introduction (reference 13)

Materials and Methods

This is well written.

Please could the authors explain their SBRT technique in more detail. Did they include patients with central or ultracentral metastases? In case of multiple lesions total dose was sometime reduced in order to preserve dose limits to organs at a sentence has been include in 3.1  Response to SBRT “ the maximum diameter we treat is  5 cm”, and the median dose was 48 Gy per nodule (IQR, 40–54 Gy).  In patients with multiple nodules each SBRT course treated 1–4 nodules maximum ( 3 all together at maximum if homolateral).. Generally in centrally located lesions total dose was reduced (40 Gy in 5 fractions) or splitted in a greater number of fractions (54 Gy in 8 fractions) to reduce the risk of late complications”.

Please could the authors clarify why they used OS as either time from metastatic event or SBRT session. Would it not effect the results to use different time points for different patients?

The primary end point was Overall Survival from first SBRT session to last visit or death but we also  calculated the overall survival from first metastatic event to last visit or follow  up reported as secondary endpoint also.This choice is due to the fact that SBRT 1st course was not usually the first treatment  at 1st Lung  Metastases  relapse   and different other treatment at different point of the patients history make overall survival from first metastasis to last follow up more related to the effect of multimodal treatment.

Results

Please could the authors address the issue of surgery beiong used more at first relapse? Is this patient selection going to effect the results and could potentially more patients benefit from SBRT than shown here if some of the younger, fitter patients had SBRT rather than surgery initially?We do not have a fixed policy of treatment and usually in multidisciplinary team we prefer  surgery for young ,fit patient and when an histology assessment is requested, but in recent years SBRT is going to be evaluated  more frequently also in younger and fitter patients.

Please could the authors explain table 3 a little more as it took me some time to make sense of it. What does the other column refer to:“Other “ Column in tab 3 refers to other therapies like Surgery+ CT  or  Wait an See,  a new label  (other TX)  was introduced in the column

Figure 2 needs labelling.resolved

Discussion

This is a good summary of the current use of SBRT for sarcoma and is well written.  Could the authors comment on the histological subtypes treated and whether they think SBRT is an option for all types of sarcoma, including those which are felt to be more inherently radioresistant such as MPNST or Triton tumours.  The number of radio resistant STS as MPNST or Triton tumor, is too small to assess if response to SBRT is different compared to more radio sensitive histology.